# Implementation of a Circular Bioeconomy: Obtaining Cellulose Fibers Derived from Portuguese Vine Pruning Residues for Heritage Conservation, Oxidized with TEMPO and Ultrasonic Treatment

Liliana Araújo [1,2], Adriana R. Machado [2,3], Sérgio Sousa [2], Óscar L. Ramos [2], Alessandra B. Ribeiro [2], Francisca Casanova [2], Manuela E. Pintado [2], Eduarda Vieira [1] and Patrícia Moreira [1,2,*]

1 Centro de Investigação em Ciência e Tecnologias das Artes (CITAR), Universidade Católica Portuguesa, 4169-005 Porto, Portugal; lparaujo@ucp.pt (L.A.); evieira@ucp.pt (E.V.)
2 Centro de Biotecnologia e Química Fina (CBQF), Universidade Católica Portuguesa, 4169-005 Porto, Portugal; armachado@ucp.pt (A.R.M.); sdcsousa2@gmail.com (S.S.); oramos@ucp.pt (Ó.L.R.); abribeiro@ucp.pt (A.B.R.); fcbastos@ucp.pt (F.C.); mpintado@ucp.pt (M.E.P.)
3 Collaborative Laboratory Towards Circular Economy (CECOLAB), 3405-155 Oliveira do Hospital, Portugal
* Correspondence: prmoreira@ucp.pt

**Abstract:** Inspired by the principles of the circular economy, using vineyard pruning residues as a source of raw materials for producing nanocellulose is a promising approach to transforming vineyard resources into value-added products. This study aimed to obtain and characterize cellulose and cellulose nanofibers from such sources. The cellulose collected from different fractions of micronized stems (500, 300, 150 μm, and retain) of vines was submitted to autohydrolysis and finally bleached. Soon, it underwent treatment via (2,2,6,6-tetrametil-piperidi-1-nil)oxil (TEMPO) oxidation and ultrasonic to obtain nanocellulose fibers. The cellulose films were obtained at a microscale thickness of 0.05 ± 0.00; 0.37 ± 0.03; 0.06 ± 0.01 e 0.030 ± 0.01 mm, with the following particle size: 500 μm, 300 μm, 150 μm, and retain (<150 μm). The bleaching efficiency of the cellulose fibers of each particle size fraction was evaluated for color through a colorimeter. In addition, the extraction of cellulose fibers was assessed by infrared with Fourier transform, and size and shape were assessed by microscopy. Differential scanning calorimetry and X-ray diffraction were performed to confirm the thermal and crystalline properties. Combining autohydrolysis with a bleaching step proved to be a promising and ecological alternative to obtain white fractions rich in cellulose. It was possible to perform the extraction of cellulose to obtain nanocellulose fibers from vine pruning residues for the development of coatings for the conservation of heritage buildings from environmental conditions through an environmentally friendly process.

**Keywords:** bleached; circular economy; nanocellulose; valorization; heritage conservation

## 1. Introduction

Cellulose has attracted considerable attention for its biocompatibility, biodegradability, and renewable properties. It is primarily derived from plants, such as trees, and offers great potential as a sustainable resource. Cellulose is an abundant organic polymer with endless natural resources. Nevertheless, its hydrophilic properties make it vulnerable to water, thus reducing its longevity [1,2].

The wine industry is a highly productive and significant sector of the agro-industry globally [3,4]. Every year, 259.9 million hl of wine are produced in the world, responsible for the generation of 20 million tons of biological residues [5]. Among the various types of residues found in the wine industry, pruning residues stand out in this study as an interesting means of understanding cellulose and account for 93% of the waste generated in viticulture [3,6]. This residue from the pruning of the vineyards is burned in the agricultural

field, causing environmental pollution. Therefore, the ecological processing and use of vine-yard pruning residues has a very important practical significance [6]. It is crucial to promote the circular bioeconomy of the winery sector by recycling and reusing their by-products to create high-value-added compounds [4]. The pruning of vineyards is a prominent source of lignocellulose, cellulose being one of the most abundant agricultural constituents.

Alternatives were explored by researchers, with various options emerging for the utilization of this residue based on its chemical composition, availability, and affordability. Through different processes, studies have evaluated its potential to produce valuable products such as hemicellulose oligosaccharides, lignin-derived antioxidants, organic acids, bioethanol from cellulose saccharification, ashes, proteins, and extracts [7–9]. Agro-industrial by-products can undergo pre-treatments, such as physical, chemical, and biological methods, to extract valuable compounds and minimize waste. This approach can enhance industrial profitability while promoting sustainable practices [3]. There are numerous methods of isolating cellulose from various sources, one of which is TEMPO oxidation, which allows native cellulose to be fully dispersed at the level of individual nanofibrils or elemental fibrils in water [10]. TEMPO oxidation is a method for oxidizing polysaccharides and introducing sodium carboxyl groups onto the surfaces of elemental cellulose fibrils. This method is one of the most promising, efficient, and energy-saving pretreatments for converting plant cellulose fibers into nanofibers [11].

This work describes our experience in obtaining and studying the properties of cellulose isolated from vine prunes of a Portuguese vineyard, thus seeking the use of this by-product to obtain new materials. In particular, the research focuses on the obtention of isolated cellulose from different particle fractions (500, 300, 150 μm and retain) and the further understanding of the effects of TEMPO oxidation, with the intent to optimize the process of extraction and defibrillation of fibers.

From a circular bioeconomy perspective, obtaining cellulose from alternate source materials for their processing is important. The goal of this research is to obtain nanocellulose fibers from residues (vine pruning) for the development of coatings for the protection of cultural built heritage. The research on the use of vine pruning to obtain and further characterize cellulose and cellulose nanofibers with respect to thickness, color, structure, and crystallinity, is important both as an innovative process for an unusual cellulose source but as well as for its novelty application, since future applications are expected to be on conservation products for built heritage.

## 2. Materials and Methods

### 2.1. Material

Vine prunings were supplied by Quinta da Lixa (Lixa, Portugal). The vine prunings were collected and dried at 60 °C for 48 h in an oven with air circulation. The dried biomass was then milled into a fine powder using a knife mill (Retsch, Haan, Germany, impact rotor mill SR 300, sieves with 0.50 mesh opening and round hole 3.00 mm) and sieved to determine the grain size distribution between 500 μm and 150 μm (500, 300, 150, and retain (fraction after 150 μm)). The milling by-products were stored until usage in sealed plastic bags to keep them from exposure to light. In this work, all fractions were used to better understand the extraction and bleaching of the by-products (Figure 1).

### 2.2. Cellulose Extraction

The cellulose fibers were extracted with the procedure described by El Halal et al. [12], with certain modifications. The material was immersed in ethanol 1:10 ($w/v$) for 16 h to remove lipids, followed by a drying process at 50 °C for 24 h. The removal of lignin and hemicellulose was performed using an alkali treatment. The vine prunes were dispersed in a 4% solution of NaOH (1:20 ($w/v$)) in an orbital shaker (120 rpm) at 80 °C for 4 h. At the end of the treatment, the solids were filtered and washed with distilled water. This alkali treatment was carried out in triplicates (Figure 1). The bleaching was carried out by adding the pruning in 1:20 ($w/v$) of buffer solution of sodium acetate (27 g of NaOH and 75 mL of

glacial acetic acid for 1 L of water) and aqueous solution of sodium chlorite (1.7%). The material was placed in an orbital shaker (120 rpm), with a buffer solution of sodium acetate at a set temperature of 60 °C for 4 h. Subsequently, the material was filtered and washed with distilled water. The bleaching process was carried out four times. The cellulose fibers were dried at 50 °C for 24 h and stored in a sealed container at room temperature (Figure 1).

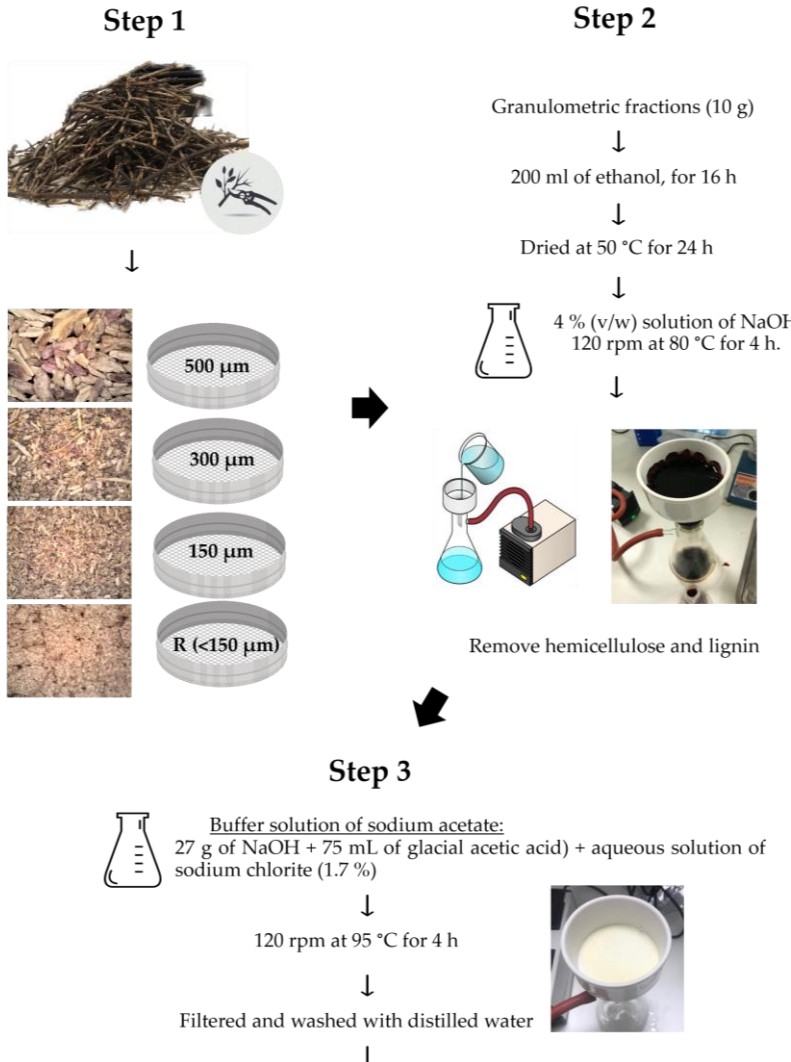

**Figure 1.** Schematic representation of the treatment steps of vine prunings: step 1—granulometric fractions; step 2—treatment; step 3—bleaching.

### 2.3. Characterization of Cellulose

2.3.1. Optical Microscopy

A high-resolution digital microscope was used to collect visual data (Dino-lite Edge AM7915MZT). The extracted cellulose was topographically characterized using a 5-megapixel camera (2592 × 1944).

2.3.2. Determination of CNCs Yield

The yield of cellulose extracted was measured and calculated according to Equation (1):

$$Yield\ (\%) = \frac{m_2}{m_1} \times 100 \tag{1}$$

where $m_2$ is the total final mass of dried cellulose after extraction (g) and $m_1$ is the initial mass (g).

### 2.3.3. Color Analysis

The color of the samples was determined using a digital colorimeter model Chroma Meter CR-700 (Konica Minolta, Osaka, Japan) using the CIELab scale to determine *L*\*, *a*\*, and *b*\* color parameters [13]. The sample was poured into a Petri dish (with 5 cm of diameter) covering the entire bottom of the dish and the reading was performed at 10 different points. The total color difference (Δ*E*) was calculated for granulates and films according to Equation (2):

$$\Delta E = \sqrt{\Delta L^2 + \Delta a^2 + \Delta b^2} \tag{2}$$

In this study, *L*\*, *a*\*, and *b*\* stand for the values of the color parameters of fresh stalk vine, and *L*\*, *a*\*, and *b*\* for the values of the color parameters of the sample in each step.

### 2.3.4. Film Thickness

The thickness of the film was determined at seven random points using an electronic micrometer (ADAMEHL—LHOMARGY, MI.21).

### 2.3.5. Water Activity

The determination of the water activity of the films was performed by direct reading using an LabMaster-aw neo, Novasina, Switzerland, operating at 25 °C. The samples were sized in a circular shape, with a diameter of 35 mm and thickness ranging from 0.120 to 0.155 mm, and placed in plastic capsules of the equipment [14].

### 2.3.6. Water Vapor Permeability (WVP)

WVP was gravimetrically determined at 25 °C by the ASTM E96-80 method, with modifications. The films were applied to permeation cells with 50 mL of distilled water inside. These cells were then packed in desiccants containing blue silica gel and monitored by weighing for seven days at 24 h intervals [14].

### 2.3.7. Fourier Transform Infrared Spectroscopy with Attenuated Total Reflectance (FTIR-ATR)

The cellulose extracted and films were confirmed by Fourier transform infrared spectroscopy (FT-IR) in on a PerkinElmer Paragon 1000 FTIR (Waltham, MA, USA) with an ATR accessory (Diamond/ZnSe). The samples were analyzed by a spectrum ranging from to 4000 to 400 cm$^{-1}$, at 4 cm$^{-1}$ resolution with 32 scans. Based on the literature, the FTIR-ATR vibrational bands were identified.

### 2.3.8. Scanning Electron Microscopy (SEM)

The morphology of cellulose fibers and film structures, such as lengths and diameters, were evaluated by SEM.

Samples were mounted on aluminum pins with double-sided adhesive carbon tape (Nisshin, Tokyo, Japan) and analyzed using a Phenom XL G2 SEM (Thermo Scientific, Eindhoven, the Netherlands). Cellulose samples were sputter coated (Polaron, Germany) with a gold/palladium alloy, but cellulose films were not, to avoid covering the roughness.

Concerning cellulose samples, SEM was operated in a high vacuum at an accelerating voltage of 5 kV and magnifications of 700× and 4000× (particle morphology assessment), and observations were performed using the secondary electrons detector (SED). Cellulose films were analyzed in a low vacuum (60 Pa), at an accelerating voltage of 15 kV, and observations were performed using the backscattered electrons detector (BSED).

### 2.3.9. X-Ray Diffraction

Powder X-ray diffraction (PXRD) analyses were performed on a Rigaku MiniFlex 600 diffractometer with Cu kα radiation, with a voltage of 40 kV and a current of 15 mA

$(3° \leq 2\theta \geq 60°$; step of 0.01; and speed rate of $3.0°/min$). The Segal crystallinity index (CI, %) was calculated using the following Equation (3):

$$CI(\%) = \frac{I_t - I_a}{I_a} \times 100 \tag{3}$$

where It is the total intensity of the (200) peak for cellulose Iβ at 22.7° 2θ and of the (020) peak for cellulose II at 21.7° 2θ, and Ia is the amorphous intensity at 18° 2θ for cellulose Iβ and at 16° 2θ for cellulose II [15,16]. All analyses were performed in duplicate.

### 2.3.10. Differential Scanning Calorimeter (DSC)

Thermal analyses were performed resorting to a differential scanning calorimeter (DSC) reference 204 F1 Phoenix® (NETZSCH-Gerätebau GmbH, Selb, Germany). Calibration of temperature and enthalpy scale was carried out using an indium/zinc standard. Samples (i.e., weight of ca. 3–6 mg) and reference (i.e., empty pan) were hermetically sealed and heated in aluminum pans over a range of 20–300 °C at a constant rate of 10 °C min$^{-1}$. Inert atmosphere was maintained by purging nitrogen gas at a flow rate of 100 mL min$^{-1}$. Each sample was analyzed in duplicate.

### *2.4. Film Production*
### 2.4.1. TEMPO Oxidation–Sonication

The procedure was followed from Saito et al., 2007, [17] with some modifications. The cellulose fibers (1 g) were suspended in a solution of TEMPO (0.016 g, 0.1 mmol) and sodium bromide (0.1 g, 1 mmol) in water (100 mL). The pH of the 12% NaClO solution was adjusted to 10 by adding 0.1 M HCl. The TEMPO-mediated oxidation was initiated by adding the necessary amount of NaClO solution (2.5 mmol NaClO per gram of cellulose), and it was maintained at room temperature by stirring at 500 rpm. By adding 0.5 M NaOH, the pH was kept at 10. The solution was sonicated at an amplitude of 60% for 10 min, then filtrated and dried for 24 h at 50 °C.

### 2.4.2. Solvent Casting

The membranes were created using a cellulose film fabrication casting procedure [18]. For NFC film preparation, a 0.52 wt% cellulose solution in deionized water was utilized. In the NFC solution, 10% glycerol by weight of cellulose was added, and the final solution was homogenized for 5 min with an ULTRA-TURRAX, at 13,000 rpm. It was then poured into a plastic dish and kept in a drying oven at 40 °C for approximately 24 h, resulting in a free-standing transparent cellulose film.

### *2.5. Statistical Analysis*

All tests were performed at least in triplicate and represented in the form of mean ± standard deviation. The results were analyzed with GraphPad Prism 8 software (GraphPad Software, Inc., San Diego, CA, USA).

## 3. Results
### *3.1. Cellulose Extraction from Vine Stalks*

The extraction of cellulose from vine trimmings was conducted with alkaline extraction (Figure 2A) and subsequent bleaching (Figure 2B) of the fiber. Figure 2 depicts the micronization treatments and yields for each granulometric fraction. The micronized vine pruning of each fraction that was initially subjected to an alkaline treatment displayed a light brown color (Figure 2B), indicating a reduction in hue when compared with the original material, as a result of the lignin remotion. Alkaline hydrolysis with sodium hydroxide allows the breakdown of the alpha-aryl ester bonds from the polyphenolic monomers, decomposing lignin while solubilizing it through hydrogen bond weakening, increasing the pulp purity in cellulose. Following that, the bleaching procedure was successful in whitening cellulose, with the use of sodium chlorite yielding the most holocellulose (hemi-

cellulose + cellulose) in a pure state. The yellow undertones that remain after the alkaline treatment are remnants of lignin and hemicellulose. The bleaching procedure with sodium chlorite aids in the elimination of these residual traces. According to the authors Trilokesh and Uppuluri, ref. [19], cellulose that displays a transition from yellow to the final white color is cellulose in high purity, whereas cellulose with a yellow-brownish tone is cellulose with impurities (lignin and hemicellulose), as illustrated in Figure 2A. Furthermore, Figure 2C details the yield of each granulometry, showing higher yields in bigger fractions and lower yields in smaller fractions, suggesting that different granulometry yields a different maximum of pure cellulose. The largest particles used for the extraction of cellulose accumulated a higher percentage of lignin, cellulose, and hemicellulose.

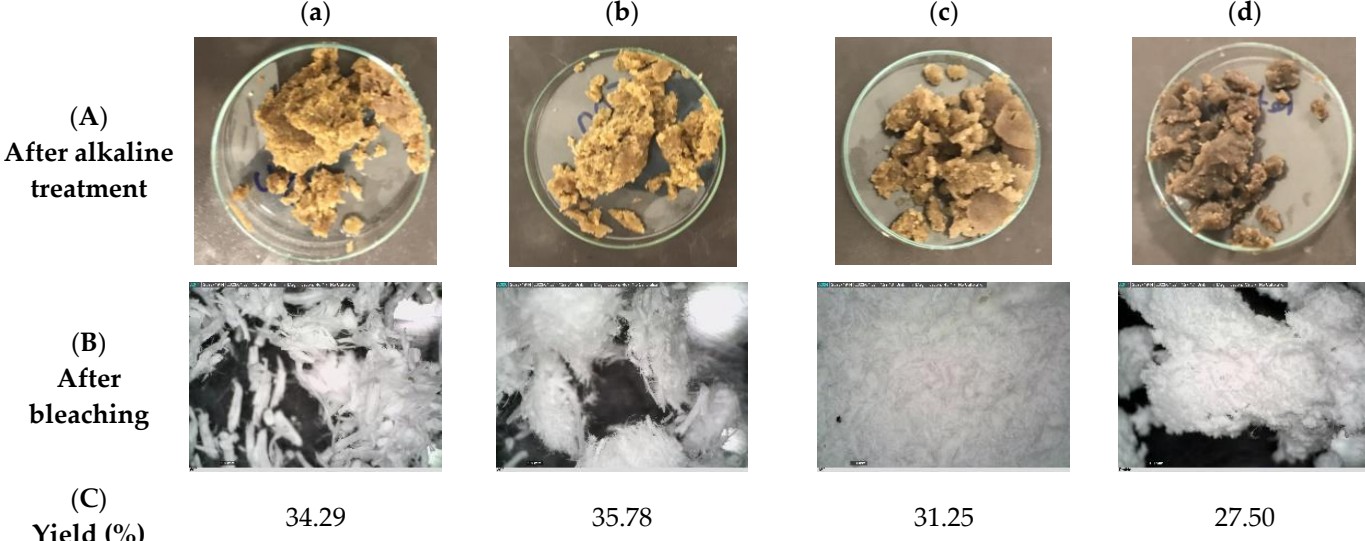

**Figure 2.** Cellulose obtained from different treatments, such as alkaline (**A**) and bleaching (**B**) processes. The cellulose from different granulometries: (**a**) 500 µm, (**b**) 300 µm, (**c**) 150 µm, and (**d**) retain (<150 µm), acquired from Dino-lite® Edge AM7915MZT. Biomass yield (%) obtained after each cellulose extraction (**C**).

Pujol and co-authors [20] reviewed the different granulometric fractions of grape stalks waste and reported that milling processing can influence chemically and structurally the final makeup of cellulose. Accordingly, the fraction of lignocellulose is accumulated in the largest fractions, while extractives and organic compounds are more likely to be present in smaller particles.

Similarly, Ratnakumar and co-authors [21] studied the impact of different particle sizes of rice straw to extract cellulose. Particles with lower sizes obtained smaller yields (27.19 ± 0.99%) while particles between 150 to 250 µm yielded more cellulose (38.31 ± 0.85%)

In sum, the fractionized particle could be utilized to enrich components by utilizing variances in chemical and structural makeup.

The cellulose fiber obtained from the extraction was characterized for its color parameters by a colorimetric analysis (Figure 3b). Plus, an analysis of the color from the original material (Figure 3a) was also performed.

CIELab analysis of untreated vine pruning from each granulometry points to a dark brown color, in Figure 3a. The darker color relates to the presence of lignin and hemicellulose as well as other extractives (waxes and lipids). Nevertheless, the color from each fraction appears to be significantly different between them. The retain (<150 µm) fraction is shown to be darker than the other fractions, while 300 µm is lighter than 150 µm. As mentioned, smaller particle sizes, such as retain, occur to have more extractives and organic compounds that could account for a darker color. Congruently, in Figure 2A, after alkaline extraction, the fragments are still darker when compared with other fractions, indicating

that the extraction treatments can be optimized to take into consideration higher amounts of extractives and organic compounds.

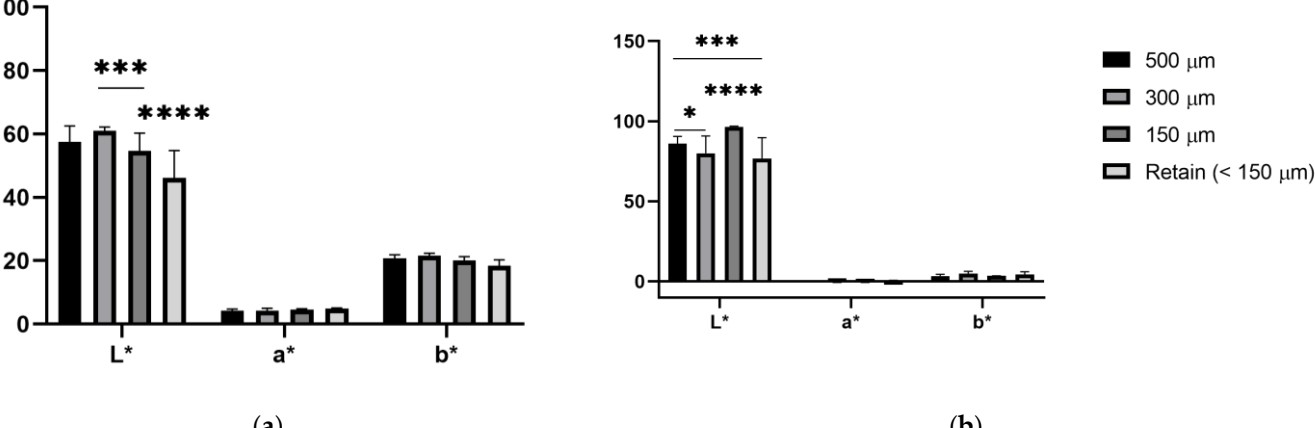

(**a**) (**b**)

**Figure 3.** Color analysis of (**a**) original material and cellulose (**b**) isolated. Error bars represent the standard deviation from ten measurements. *, $p < 0.05$; ***, $p < 0.001$, ****, $p < 0.0001$.

Initiating the extraction process with the immersion of the material in ethanol helps with the extraction of cellulose by removing lipids and waxes. Consequently, the arrangement of alkaline and bleaching treatment with ethanol guarantees the extraction of cellulose by stripping off lignin, hemicellulose, and all the remaining extractives. The extracted cellulose shows a whiter color, confirming that the cellulose obtained is in a pure state. Consequently, different tons of white are displayed, and the 150 μm fraction is significantly lighter than the other fractions. The color analysis could indicate that the treatment is more effective in the 150 μm fraction as a whitened purer cellulose is obtained. In a previous study, cellulose was extracted from vine stalks and bleached with hydrogen peroxide. The bleached cellulose presented a yellow tone indicating that the bleaching process was not efficient and traces of lignin and hemicellulose still remained [22].

*3.2. Fourier Transform Infrared (FTIR) Spectroscopy Analysis of Isolated Cellulose from Vine Stalks*

Pure FT-IR spectra of vine stalks and extracted cellulose were logged from 4000–600 cm$^{-1}$. The results were then compared with the two commercial controls, pure lignin (Sigma, New York, NY, USA) and microcrystalline cellulose (Avicel® PH-101, Sigma). In addition, spectral data for each granulometry were collected, as shown in Figure 4.

Cellulose in FT-IR spectra can be attributed to the absorption band at 3327 cm$^{-1}$, suggesting the stretching of hydroxyl groups. Furthermore, the C-H group in the glucose unit is allocated to the bands at 2900 and 1306 cm$^{-1}$ for stretching and deformation vibrations, respectively. The absorption band at 889 cm$^{-1}$ characterizes the β-glycosidic link between the glucose units [23]. Moreover, the signal at 1024 cm$^{-1}$ is attributed to the cellulose chain backbone's -C-O group of secondary alcohols and ether functionalities. Finally, the C-O-C vibration has a diagnostic peak for cellulose at 1200 cm$^{-1}$ [24].

Firstly, a spectrum for lignin (a) was determined in order to compare with the original material and extracted cellulose. The peaks observed at 1504 and 1238 cm$^{-1}$ are typically formed by stretching vibration of C=C of the aromatic rings of lignin. However, hemicellulose has a complex transmission pattern in peaks between 1200 and 900 cm$^{-1}$ [25].

The spectra obtained from the extracted cellulose appears to have a more defined shape, while the original material (Figure 4B) seems to have lower intensity. The samples in the same category (original material or extracted cellulose) show peaks in the same regions (wavenumber) only with different peak intensities.

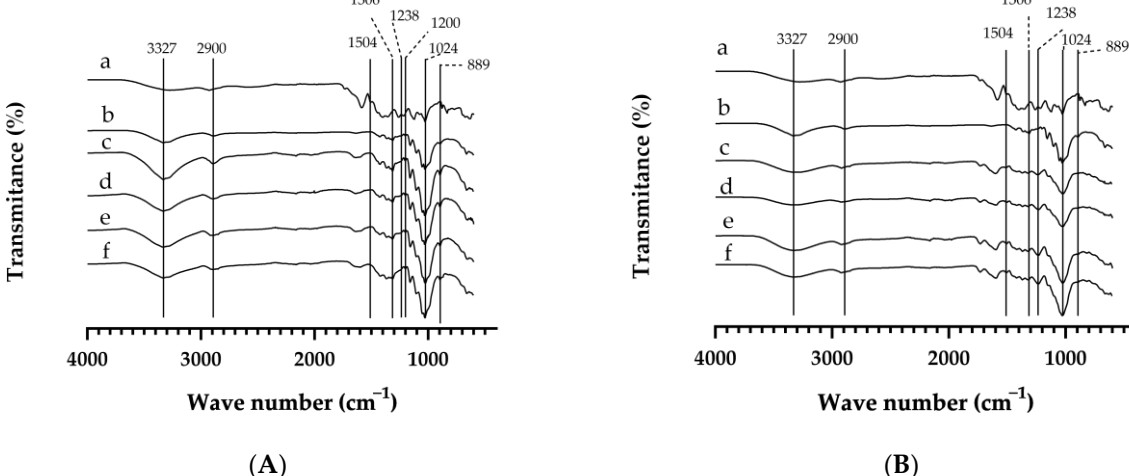

**Figure 4.** FT-IR spectra from the cellulose extracted from vine pruning (**A**) and from pure vine pruning (**B**). Spectra of (a) lignin (control), (b) microcrystalline cellulose (control), (c) 500 μm, (d) 300 μm, (e) 150 μm, and (f) retain (<150 μm).

Analyzing the spectra of the original material and extracted cellulose, the original appears to show peaks characteristic of lignin at 1504 and 1238 cm$^{-1}$ [26]. These peaks are characteristic of C=C stretching of aromatic rings in lignin. However, the spectra of extracted cellulose show an absence of these peaks, possibly indicating the removal of lignin. Despite the fact that the spectra included peaks at 2900 cm$^{-1}$, assigned to the C-H group in the glucose unit and characteristic of cellulose, this peak may also reflect the interlinkage of cellulose to lignin molecules [27,28]. The absence of peaks at 1504 and 1238 cm$^{-1}$ indicates that the alkaline treatment and the bleaching process could be effective for the removal of lignin; then again, the reduction of intensity at 2900 cm$^{-1}$, a known peak to lignin, could conclude that the treatments did not efficiently remove all the lignin constituents [23,27].

The polysaccharide group known as hemicellulose has a complex absorption pattern, with the primary distinguishing peak falling between 1200 and 900 cm$^{-1}$.

The bands characteristic of cellulose, such as 889, 1024, 1200, 1306, and 3327 cm$^{-1}$, appear to intensify in the spectra of extracted cellulose (Figure 4A), suggesting that the samples were more pure after alkaline and bleaching treatments. Additionally, it was determined that the peak at 1632 cm$^{-1}$ was caused by O-H-O bending vibration in absorbed water.

### 3.3. Isolated Cellulose Fibers SEM Analysis

The bleached fibers showed an individualized and fibrous structure, with a rod cell and an elongated shape. The different granulometric fractions displayed fibers of various sizes. The fractions of 500 and 300 μm (Figure 5a,b) showed thicker and elongated fibers, whereas the fraction of 150 μm (Figure 5c) presented thinner and aggregated fibers while maintaining the elongated form. Retain (Figure 5d), on the other hand, comprised extremely small fibers that were highly agglomerated with one another and presented an irregular shape.

### 3.4. Production of Films from Isolated Cellulose from Vine Stalks

Since the aesthetics of the films are crucial for the foreseen application and as the cellulose fiber formed at this time is white and opaque, it is necessary to fibrillate the cellulose fiber to smaller fibers and diameters. The oxidation TEMPO approach yields a more transparent membrane; nevertheless, the outcomes mediated by oxidation TEMPO generated a barely transparent membrane (Figure 6). The results showed that membranes made from smaller granulometry (retain) are more transparent. This may imply that

the oxidation process produced a clear membrane even though the reaction period was insufficient, as a smaller granulometry may take less time to obtain a more fibrillated fiber.

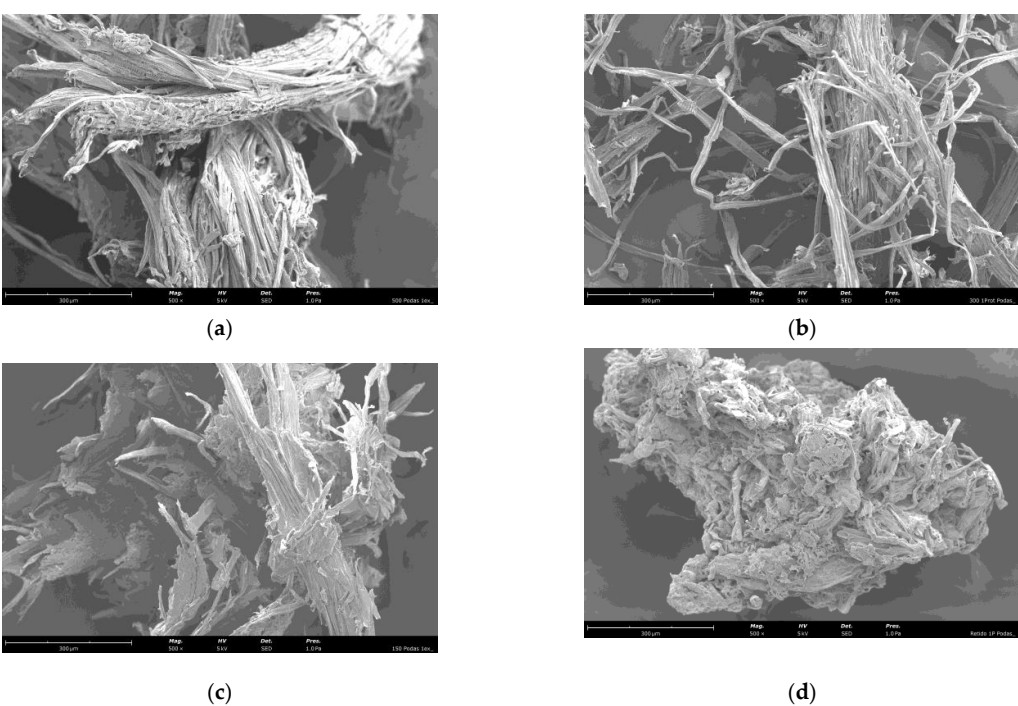

**Figure 5.** Representative SEM images of cellulose extracted from different granulometries: (**a**) 500 μm, (**b**) 300 μm, (**c**) 150 μm, and (**d**) retain (<150 μm). The scale bar represents 300 μm.

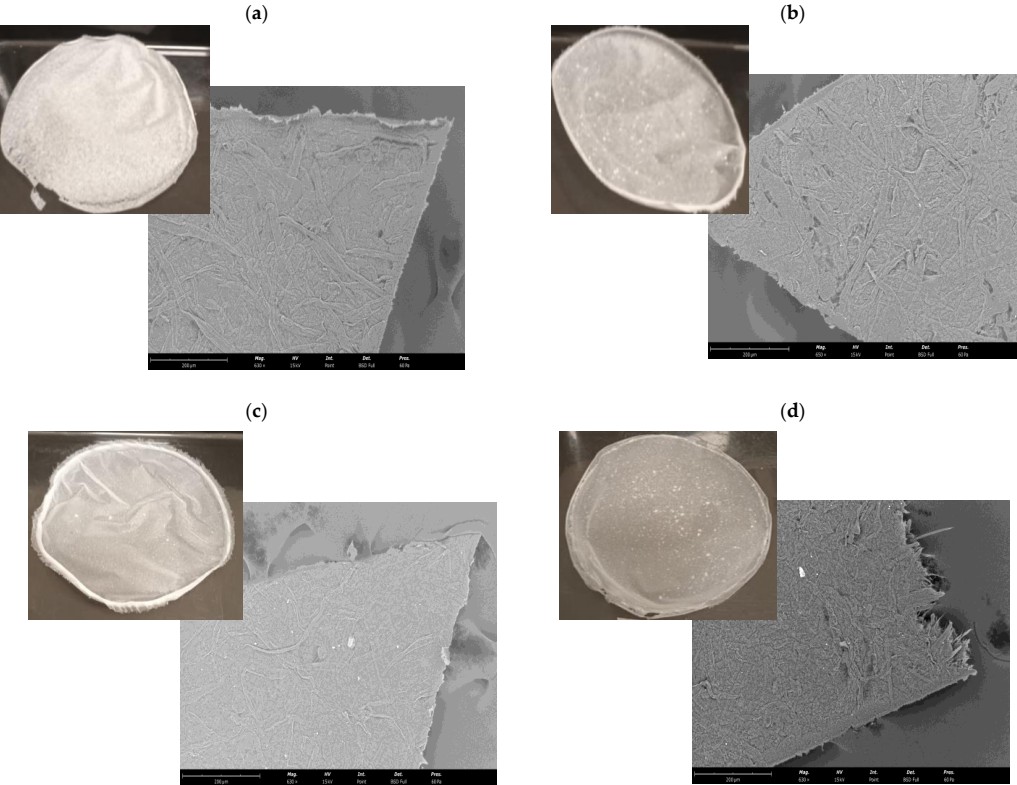

**Figure 6.** Representative photographs and SEM images of the films obtained from cellulose extracted from vine pruning of different granulometries ((**a**) 500 μm, (**b**) 300 μm, (**c**) 150 μm, and (**d**) retain (<150 μm)). Scale bars represent 200 μm.

The morphological images of the prepared films samples were obtained by SEM (Figure 6). The films exhibited a relatively homogeneous and compact structure (Figure 6). Moreover, the film obtained from the retain (<150 μm) presented a particulate morphology, which is in accordance with the one previously described regarding the morphology of those cellulose fibers.

### 3.4.1. Film Thickness

The results obtained (Figure 6) in the determination of the thickness of the films, (a) 500 μm, (b) 300 μm, (c) 150 μm, and (d) retain (<150 μm), were, respectively: $0.05 \pm 0.00$; $0.37 \pm 0.03$; $0.06 \pm 0.01$ e $0.030 \pm 0.01$ mm.

Thickness values were used to obtain measurements of mechanical properties. The thickness control of films is crucial to evaluate their uniformity and repeatability for properties measurement. Additionally, it validates comparisons between films and provides information about mechanical strength and water vapor barrier properties [29]. The films' thickness showed that the most crucial factor impacting film thickness is initial particle size, as the film becomes more thinner as the initial particles are smaller. With the exception for the 300 μm granulometry, film thickness measurements could have been affected by an uneven film surface when higher granulometries of cellulose were used. The thickness of the films has been directly related to final particle size of the cellulose, as films produced from nanocellulose have nanoscale thickness (1–100 nm), presenting a more transparent and smooth appearance [30].

### 3.4.2. $A_w$—Water Activity, WVP—Permeability, and Color Analysis of Films

The water activity was determined by operating at a temperature of 25 °C, shown in Figure 7. A 35 mm circular form was used to size the samples. Since water activity is connected to free water in a product, items with high water activity exchange water with the environment more quickly, increasing water loss and degradation. The results from studies of water activity indicate that it can have a direct influence on their mechanical properties, since the amount of free water is not affected, maintaining the structure of the fibers, and therefore avoiding large variations with changes in relative humidity.

As a result, water vapor permeability (WVP) (Figure 7) is an important metric for the characterization and applicability of the films. The films with glycerol inclusion were thicker, with enhanced permeability to water vapor proven by the presence of pores in their structure and/or increased thickness, probably due to the more open structure. The plasticizer reduces the number of hydrogen bonds between polymer chains and interferes with the molecular space [31]. Therefore, the addition of glycerol, a renewable plasticizer, to biodegradable films improves malleability and properties useful for various applications [32].

Obtaining films with good water vapor barrier properties, that is, with low permeability within a large range of relative humidity, implies the use of insoluble material or low solubility in water, but this does not disqualify the film, which will depend on the place where it will be applied [14].

By containing the parameters obtained in the colorimetry test, it was possible to arrange the results shown in Figure 7. The staining and opacity that the films may present are a result of the chemical or morphological structure of the macromolecule used; the type of solvent can also influence the final color of the polymer matrix [33,34].

Among the choice of materials used for the production of films, the plasticizer is of great importance, because it must be compatible with the choice of solvent and polymer used, which presents interaction with the macromolecule so that separation does not occur during drying [33].

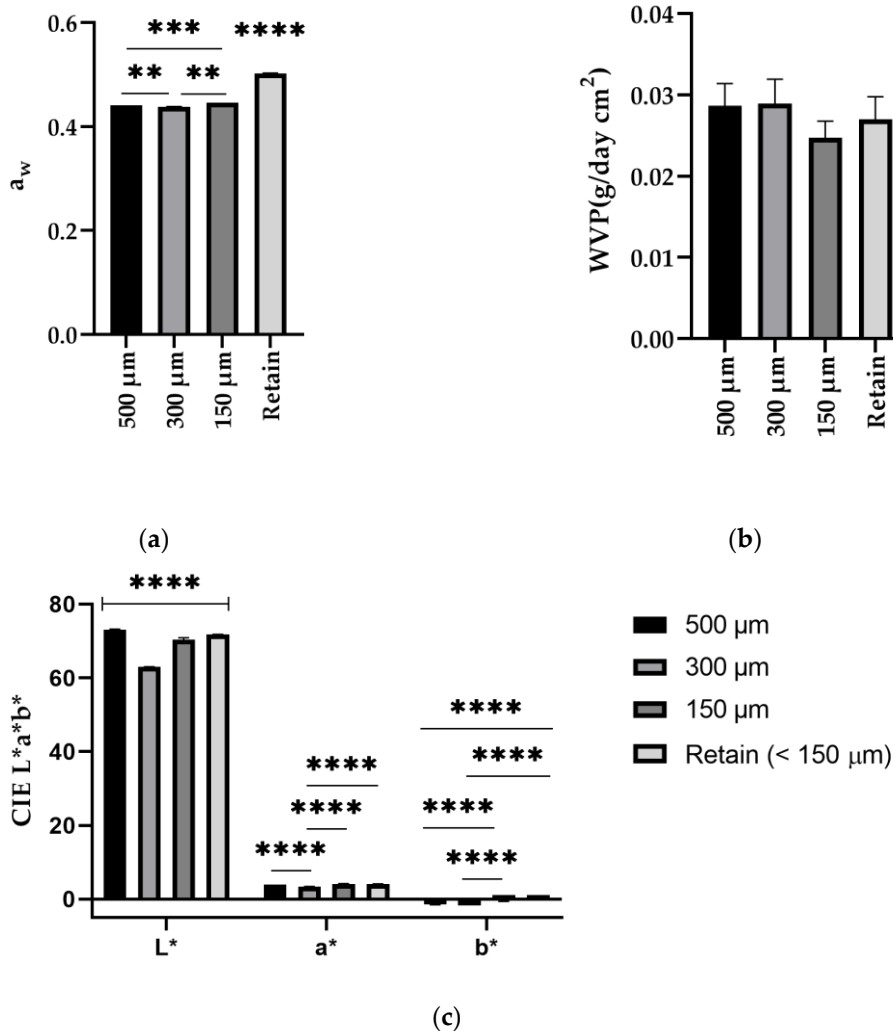

**Figure 7.** Performance of cellulose membranes made from vine pruning: (**a**) water activity, (**b**) water vapor permeability, and (**c**) CIELAB from films 500 μm, 300 μm, 150 μm, and retain (<150 μm). Error bars represent the standard deviation from ten measurements. **, $p < 0.01$, ***, $p < 0.001$, ****, $p < 0.0001$.

Plasticizers are used to improve aspects such as flexibility, mechanical strength, and adhesiveness, as well as to reduce discontinuous and brittle zones. The amount of plasticizer to be added to the filmogenic solution is extremely important because it will directly affect the properties of the films. In addition, when the amount added is excessive, it can exude during drying, despite having an affinity with the mixture [33,34].

The color of the films (Figure 7c) shows the difference between size fractions; the L* values were significantly different in all fractions. The lightness of all film was different for different particle sizes; however, the transparent nature of the film made it difficult to obtain an accurate measurement of color. The value of a* and b* show some significant differences. Conversely, the transparent film is hard to accurately evaluate this value. The film 150 μm and retain appear to be more transparent and have a more yellow undertone (+b* value) and a red tone (+a* value). Additionally, in Table 1 is tabulated the ΔE calculated between granulates and films after the oxidation TEMPO. Except for the retain, the difference of color appears to increase with lowering the granulometries. In 150 μm fractions, these values are in accordance with the visual observation of the higher transparency of the film formed. The film made from the retain fraction has the lower difference, which can indicate that the fiber was subjected to lower alterations [35,36].

**Table 1.** Calculated ΔE difference between granulates and films color.

|  | ΔE |
| --- | --- |
| 500 μm | 14.41 ± 4.51 |
| 300 μm | 18.77 ± 10.82 |
| 150 μm | 26.81 ± 0.34 |
| Retain (<150 μm) | 7.75 ± 12.97 |

As previously stated, the oxidation TEMPO technique produces a more transparent membrane, and yet the results mediated by oxidation TEMPO produced a scarcely transparent membrane. According to the findings, membranes constructed of smaller granulometries (150 μm and retain) are more transparent.

*3.5. FTIR Analysis of Produced Films*

From the FTIR spectrum, it is possible to compare the characteristics of the controls (lignin and microcrystalline cellulose) and those of the films produced and identify the main chemical changes that occurred during their production. The spectra referring to the cellulose films obtained through different granulometries (c, d, e, and f) are presented (Figure 8). Based on the spectra obtained, it is possible to verify common absorption bands for both samples. Both materials produced in this study, display the presence of a characteristic peak in 3327 cm$^{-1}$ attributed to the elongation of the OH of cellulose, showing greater intensity for the cellulose films with the different granulometries (c, d, e, and f). Both films produced showed sharp peaks in the regions of approximately 2900 cm$^{-1}$, which was determined to be the result of stretching; anti-symmetric and symmetrical vibrations of groups -CH$_2$, and 1024 cm$^{-1}$ corresponded to the stretching of the C-O bond in cellulose and in the hydroglucose group [37,38]. The peak 1624 cm$^{-1}$ is related to the OH of water absorbed from cellulose. In comparison with cellulose extracted (Figure 4A), the spectra show a very similar profile to the film spectrum (Figure 8). However, the peak 2900 cm$^{-1}$ correlated to cellulose linked to lignin shows different intensity. The film of 500 μm presents a more intense peak, which could indicate that the fibrillation of cellulose into smaller fibers increases the exposition of traces of lignin. The process of defibrillation of larger particles could expose traces of lignin still inside the fiber.

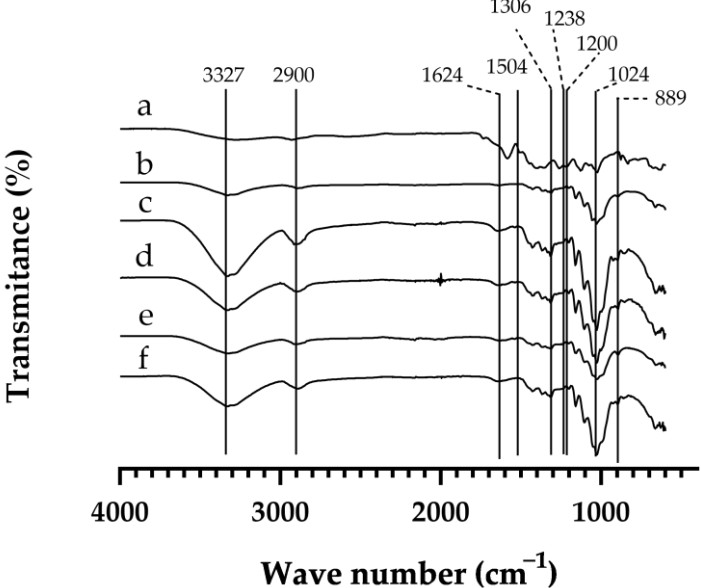

**Figure 8.** FTIR spectra of (a) lignin (control), (b) microcrystalline cellulose (control), films (c) 500 μm, (d) 300 μm, (e) 150 μm, and (f) retain (<150 μm).

The FTIR spectra of the control (microcrystalline cellulose) (Figure 8) presents less intense peaks than the films. In FTIR, differences in peak intensity indicate differences in the amount of the functional group linked to the molecular bond (per unit volume). However, the spectra from the isolated granulates, shown in Figure 4A, have similar intensities to the peaks from the controls, indicating that the molecular structure is very similar to micro-crystalline cellulose. The change in peak intensity in Figure 8 could indicate that ultrasonic and TEMPO oxidation treatments may breakdown the intra- and intermolecular hydrogen bonding between large cellulose units, resulting in cellulose in a nanoscale form, which might explain increases in peaks at 889, 1024, 1200, 1306, and 3327 $cm^{-1}$, probably related to structural alterations in cellulose after treatments. The peak 1624 $cm^{-1}$, as previously stated, is the single bending of -OH to adsorbed water and shows an intensification in Figure 8. Authors Qu et al., 2021, show overlain and increased peaks to approximately 1624 $cm^{-1}$ formed by TEMPO oxidation that was assigned to the asymmetric stretching vibration of the carboxylate group [39]. Overall, it is possible that the films, after TEMPO oxidation and ultrasound application, had an increased moisture absorption and the fibers suffer partial breakdown into smaller particles, making for a more smooth and opaque film.

*3.6. X-Ray Diffraction of the Films and Isolated Cellulose Granulates*

The crystallinity of cellulose and films were analyzed by powdered X-ray diffraction (PXRD). PXRD patterns and crystallinity index are shown in Figure 9 and Table 1.

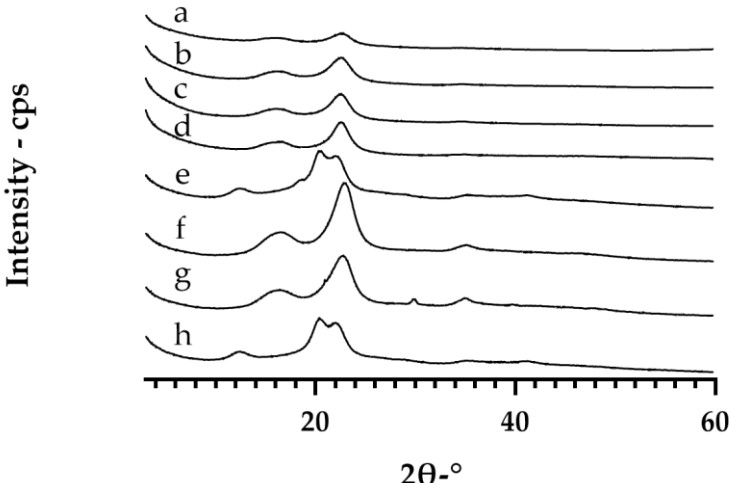

**Figure 9.** X-ray diffraction of films and cellulose of different particle size fractions. The analysis of films: (a) 500 μm, (b) 300 μm, (c), 150 μm, (d) retain (<150 μm) and cellulose granulates: (e) 500 μm, (f) 300 μm, (g) 150 μm, (h) retain (<150 μm).

To determine the crystalline–amorphous state of materials, a PXRD is used. Typically, sharp peaks indicate a crystalline material, while broad and undefined peaks indicate an amorphous material [40–42]. It is crucial to measure the crystallinity of cellulose to assess how it correlates with the resulting properties after polymorphic conversion or nanofabrication [16].

The results indicate that the crystallinity index affects the physical properties of cellulose, as shown in Table 2. The cellulose granulates have a higher crystallinity (CI ≈ 69) than the cellulosic films (CI ≈ 63), indicating that the oxidation TEMPO reaction can also influence the crystalline regions of cellulose. The oxidation TEMPO process leads to a variety of outcomes in cellulose fibers. The negative charges that the carboxyl groups produce cause repulsive forces between the cell walls in the fibers, decreasing the cohesiveness that is retained by hydrogen bonds. Furthermore, oxidation causes the fibers to hydrate and stretch, making them more flexible and allowing access to the crystalline zone [43,44].

**Table 2.** Crystallinity index of films and cellulose.

| Sample (μm) | | CI (%) |
|---|---|---|
| Film | 500 | 67.38 ± 3.24 |
| | 300 | 63.63 ± 2.99 |
| | 150 | 59.32 ± 0.61 |
| | Retain | 63.64 ± 1.57 |
| Cellulose granulates | 500 | 74.23 ± 1.72 |
| | 300 | 68.34 ± 2.92 |
| | 150 | 68.60 ± 0.63 |
| | Retain | 66.82 ± 1.85 |

CI, crystallinity index.

The more accessible crystalline zone uncovered hydroxymethyl groups on the surface of the cellulose crystals, as well as some groups located inside of the cellulose, engaged in the oxidation TEMPO process. This might reduce the crystallinity of TEMPO-oxidized cellulose by reacting with amorphous areas of the cellulose, resulting in lower crystallinities [45,46]. The results also imply that the larger the particle, the more crystalline it is, as the crystallinity index decreases with the decrease of cellulose size. This is because micronization or defibrillation treatments expose more of the amorphous structure of the cellulose by breaking the fibers into smaller pieces. Since most intra- and intermolecular interactions for the highly ordered cellulose are caused by hydrogen bonds, a lower crystallinity index is correlated to micronization or defibrillation treatments. These processes eliminate the long range and strong intra-chain (C H···O) of hydrogen bonds that are crucial for the crystalline structure of cellulose, resulting in more amorphous cellulose [47–49].

From the diffraction patterns, it can be inferred that samples F and G present a type II cellulose structure, while the others present a type I cellulose structure. Therefore, the X-ray diffractograms obtained (Figure 9) are typical of cellulose I, the most widespread crystalline form of the four existing cellulose polymorphs, with well-defined crystalline peaks at approximately 22° and 25° [16,50]. Compared with the study by Freixo et al. [51], with sugarcane bagasse to obtain cellulose, we verified the highest crystallinity index of 69.3% using the treatment described as acid hydrolysis 3 (HA3), which is comprised of two sequential hydrolysis steps: alkaline hydrolysis with sodium hydroxide followed by acid hydrolysis with hydrochloric acid. Thus, it is important to use the treatments to obtain an adequate crystallinity for cellulose and the removal of lignin and hemicellulose components.

### 3.7. DSC of the Films and Isolated Cellulose Granulates

The cellulose extracted was analyzed by DCS; the results (Figure 10) showed for both cellulose films and granulates, Figure 10A and 10B, respectively, one dominant and evident endothermic peak, between ca. 100 °C and 146 °C, which is attributed to moisture evaporation. For cellulose films (Figure 10A), it is possible to observe a considerable shift for higher temperatures for film composed by retain (<150 μm), with the highest enthalpy (i.e., −235.95 j/g). This is likely due to a higher content of intrinsic bonded water which may be related to the lower size of particles composing these films. It is well known that the surface area to volume ratio for smaller particles is higher, and so the hydroxyl groups of cellulose will be more exposed to water molecules and therefore contribute to higher water absorption. This is likely the reason why a higher energy (enthalpy) is needed to promote the evaporation of water molecules which are intrinsically bounded. For cellulose granulates (Figure 10B), the moisture content is lower, which is possible to see by the lower intensity of the endothermic peaks, but also by the lower energy (enthalpy) needed to evaporate the bounded water in comparison with cellulose films. The presence of water in the cellulose fibers can break down the OH-bond within the cellulose and make available these groups for interactions with water molecules. Ultimately, the water that is adsorbed into the film causes a change in the material dimensions as well as a reduction in mechanical

qualities. As fibers swell, shear pressures develop at the fiber–matrix interface, which leads to the deterioration of the fiber–matrix interface area [52].

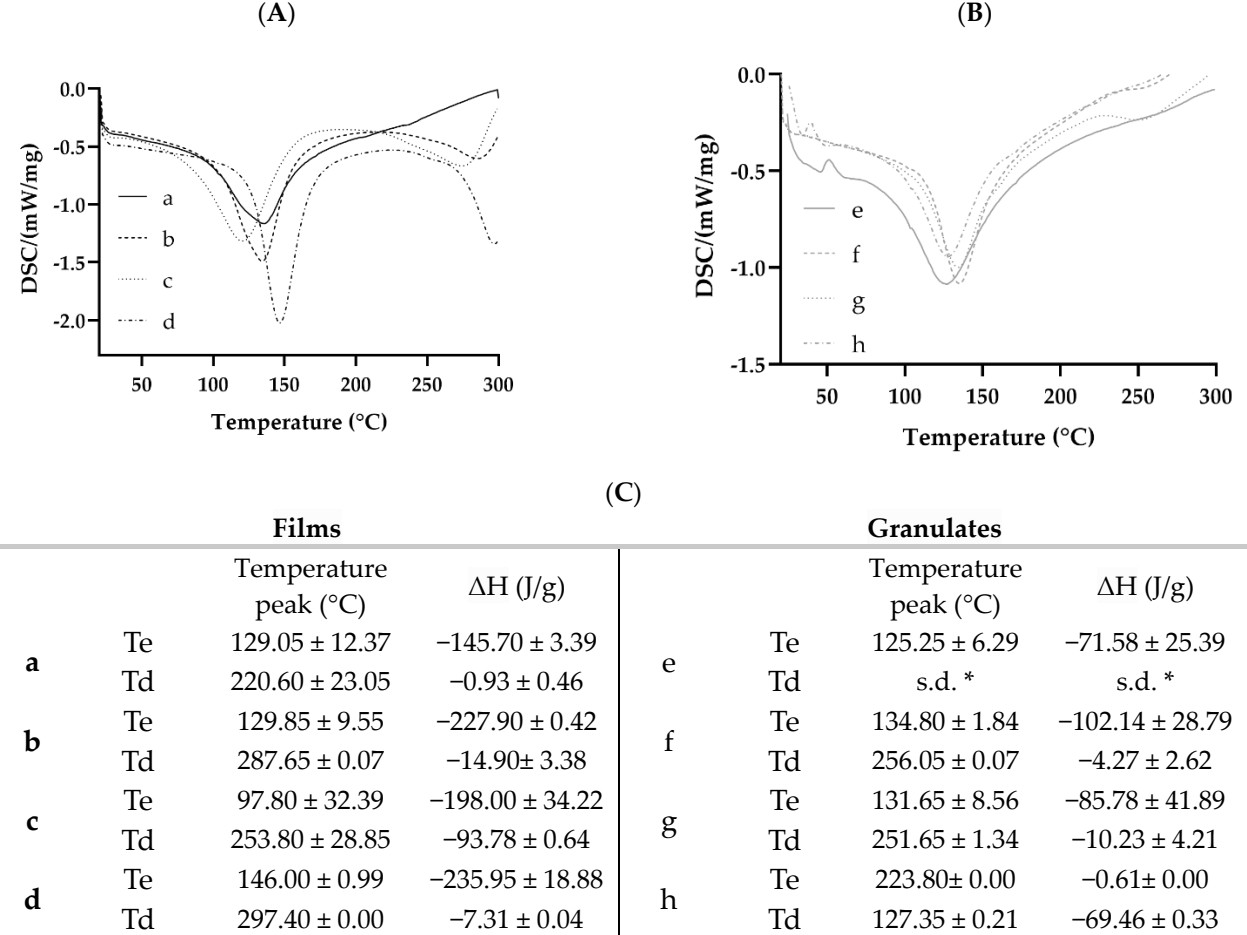

**Figure 10.** DSC graphs of films (**A**) and cellulose granulates (**B**) of different particle size fractions. The analysis of DSC results of cellulose films: (a) 500 μm, (b) 300 μm, (c). 150 μm, (d) retain (<150 μm) and cellulose granulates: (e) 500 μm, (f) 300 μm, (g) 150 μm, (h) retain (<150 μm). The main temperature peaks and enthalpy of degradation (ΔH) (**C**). * Not detectable up to 300 °C (Te—temperature of evaporation; Td—temperature of decomposition).

From Figure 10, it is also possible to observe for some samples the presence of a second endothermic peak attributed to the decomposition of cellulose. For films (b), (c), and (d) with 300 μm, 150 μm, and retain (<150 μm) fractions the cellulose started to decompose at ca. 288, 254, and 297 °C, respectively. For films with 500 μm fraction, it is possible to see a glass transition at approximately 220 °C, which indicates that a rearrangement of interactions between cellulose chains may be occurring caused by water evaporation. The decomposition of cellulose was not possible to observe for this sample because it may be occurring at temperatures higher than 300 °C, demonstrating that it may be more stable than from the other fractions.

Analyzing the cellulose granulates (Figure 10B), it is possible to observe a second endothermic peak corresponding to the decomposition of cellulose at temperatures of ca. 256 °C and 252 °C for samples (f) and (g) with 300 μm and 150 μm fractions, respectively. For the remaining granulates, it is not possible to visualize any additional endothermic peak for the decomposition of cellulose, thus suggesting that it is occurring at temperatures higher than 300 °C.

The secondary endothermic peak present at 300 °C was attributed to the melting of cellulose, which corresponded to the disintegration of glycosidic linkages, and cellulose depolymerization [53].

The temperature and intensity of the secondary peak can provide information about the thermal stability of the cellulose substance. The main peak may correspond to the initial, more easily degradable cellulose fractions, while the secondary peak, present at higher temperatures, indicates that a more stable cellulose component is undergoing decomposition [54].

The significance of a secondary peak can vary depending on the intended use of cellulose-based materials. For instance, understanding the secondary peak might assist in optimizing processing conditions to prevent degradation during manufacturing of cellulose-based films or coatings. Understanding the steps of decomposition is essential for process control in situations where cellulose degradation is either required or needs to be absent.

## 4. Conclusions

The extraction of cellulose from vine pruning is a promising strategy to obtain these polymers from alternative sustainable sources. The methods utilized in this study can extract cellulose to be used for film development for innovative applications for the conservation of heritage buildings. The use of agro-industrial waste can provide several advantages such as reducing dependence on non-renewable sources to obtain cellulose (as wood) as well as waste reduction, lowering environmental impact, and allowing for the production of other products with added value.

The structural and morphological characteristics of the produced cellulose and films were investigated by FTIR, X-ray diffraction (XRD), and scanning electron microscopy (SEM). Surface morphology, material structure, molecular structure, a microcrystalline structure, and color difference values were measured before and after the film, showing higher transparency of the end product. The XRD measurements verified the crystalline characteristics. The final film obtained had nanofibers from the oxidation TEMPO obtained from the original micro cellulose.

However, the extraction of cellulose from agro-industrial waste, specifically vine pruning, might present technical obstacles such as cost and material availability, as vine pruning is only generated once a year and the process is difficult to scale up. Choosing the optimal extraction method for each kind of waste, optimizing operating conditions, recovering and reusing old solvents and reagents, lowering energy and water consumption, as well as limiting effluent and emission production are all part of the process. As a result, more research and innovation are needed to develop efficient, cost-effective, and environmentally acceptable cellulose extraction processes from agro-industrial waste. Furthermore, the application of cellulose films in cultural heritage settings is also within the future steps of this research.

**Author Contributions:** Conceptualization, L.A., A.R.M. and P.M.; methodology, L.A. and A.R.M.; validation, P.M.; investigation, L.A., A.R.M., Ó.L.R., S.S., A.B.R. and F.C.; resources, L.A., Ó.L.R., S.S. and A.R.M.; data curation, L.A., Ó.L.R., S.S. and A.R.M.; writing—original draft preparation, A.R.M. and P.M.; writing—review and editing, L.A. and A.R.M.; supervision, P.M., E.V. and M.E.P.; funding acquisition, P.M. and M.E.P. All authors have read and agreed to the published version of the manuscript.

**Funding:** Adriana R. Machado thanks their research contract funded by Fundação para a Ciência e Tecnologia (FCT) and project CENTRO-04-3559-FSE-000095—Centro Portugal Regional Operational Program (Centro2020), under the PORTUGAL 2020 Partnership Agreement, through the European Regional Development Fund (ERDF). The authors acknowledge the financial help from project HAC4CG—Heritage, Art, Creation for Climate Change. Living in the city: catalyzing spaces for learning, creation, and action towards climate change. NORTE-45-2020-75. SISTEMA DE APOIO À INVESTIGAÇÃO CIENTÍFICA E TECNOLÓGICA—"PROJETOS ESTRUTURADOS DE I&D&I" HORIZONTE EUROPA.

**Institutional Review Board Statement:** Not applicable.

**Data Availability Statement:** Not applicable.

**Acknowledgments:** The authors thank CITAR, CECOLAB, and CBQF, Portugal.

**Conflicts of Interest:** The authors declare no conflict of interest.

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
