# Peer review of "Implementation of a Circular Bioeconomy: Obtaining Cellulose Fibers Derived from Portuguese Vine Pruning Residues for Heritage Conservation, Oxidized with TEMPO and Ultrasonic Treatment"

_agriculture, doi:10.3390/agriculture13101905_

Round 1

Reviewer 1 Report

The manuscript entitled "Implementation of a Circular Bioeconomy: Obtaining cellulose fibres derived from Portuguese vine pruning residues for Heritage conservation, oxidized with TEMPO and ultrasonic treatment". The cellulose and cellulose nanofibers were characterized by multiple characterization equipments. The thermal and crystalline properties of as-prepared samples have been studied. Related comparative experiments were also performed. The experimental work is interesting. In my opinion, this work is interesting and has a certain reference in the development for the application in the fields of coatings for the conservation of heritage buildings. However, there are some remarks that should be taken into consideration by the authors in order to raise this article to a good level for publication in Agriculture.

The suggested modifications are listed as follows:

1. Acronyms that first appear should be explained in detail. Such as: TEMPO.

2. All formulas should be numbered sequentially.

3. The text in Figure 2(2) is not clear and should be reprocessed.

4. Is there only 3 a* values in Figure 3(b)? a* value is negative, the sample contains a small amount of green component?

5. In Figure 4, the characteristic peak at 1632 cm -1 can be ascribed to the absorbed water. Some literature can be used for reference: [1] https://doi.org/10.1016/j.nucana.2022.100026; [2] https://doi.org/10.1016/j.ceja.2021.100089.

6. The text in Figure 5 and 6 is not clear and should be reprocessed.

Author Response

The authors would like to thank the reviewers for the suggestions made, we have carefully replied to all reviewer’s comments as seen below. All changes in the manuscript text have been highlighted in green.

Reviewer 1

Answer:

  1. The suggestion was taken into consideration and the text was modified accordingly.
  2. The suggestion was taken into consideration and the text was modified accordingly.
  3. The suggestion was taken into consideration and the text was modified accordingly.
  4. Yes, in the graphs the sample retain has a value negative, however with the addition of the other value, the colour is in a similar range to the other colours (whitish colour).
  5. The recommendation was taken into consideration and the text was modified accordingly.
  6. The suggestion was taken into consideration and the text was modified accordingly.

Reviewer 2 Report

A work of high interest is presented in which different experimental techniques are applied. The results shown are interesting but some modifications are proposed to improve the quality of the results.

The first doubt that arises for me is the proposed application in terms of heritage conservation. This application is mentioned in the introduction and in the conclusions, but in the results analysis section it is not mentioned at any time. That is why it is not known whether the properties of these films are suitable for such an application.

The formula with which the total color difference is calculated is shown, but I have not found the results in the results analysis. Additionally I recommend to review the work 10.1002/mame.202100196 in which in the experimental section is shown to affect such parameters to the appreciation of the observer according to his experience looking for color differences.

The size scales of the sem figures are not observed correctly. It is recommended to modify the figures so that they can be correctly observed. 

To improve the identification of the results shown, it would be interesting to include in the titles of each of the sections whether we are working with the film or with the by-product. At the moment they are differentiated by numbering, but this could improve their identification.

It is proposed that there is a difference between the different film thicknesses obtained as a function of particle size. But the reason for the change is not discussed.

In different sections it is proposed that there are different parameters that affect the mechanical properties but they are not measured in this work. It would be interesting to incorporate this study as it can provide relevant information for its application.

In general terms the manuscript is well understood and can be published in the current form in this regard. But there are some errors to be revised as for example in line 463 that should be revised.

Author Response

The authors would like to thank the reviewers for the suggestions made, we have carefully replied to all reviewer’s comments as seen below. All changes in the manuscript text have been highlighted in green.

Reviewer 2

Q1. The first doubt that arises for me is the proposed application in terms of heritage conservation. This application is mentioned in the introduction and in the conclusions, but in the results analysis section it is not mentioned at any time. That is why it is not known whether the properties of these films are suitable for such an application.

Answer: The objective of the article is to explore the efficacy of the methods used to obtain cellulose from by-products and the possibility to prepare films to apply to heritage (from that cellulose). The testing of the films will be only further detailed in the next articles.  However, although the final application is not analysed in this article, the authors think it is important to explain that these results are part of a bigger project related to heritage as this is an area where innovation regarding by-products and circular economy is less explored.

Q2. The formula with which the total colour difference is calculated is shown, but I have not found the results in the results analysis. Additionally, I recommend to review the work10.1002/mame.202100196 in which in the experimental section is shown to affect such parameters to the appreciation of the observer according to his experience looking for Colo differences.

Answer: Initially, the authors planned to compare the original material with the isolated cellulose, but didn’t make sense. We thank the reviewer for pinpointing this error. We further compared between the granulates and the films, as it is important in this case.

Q3. The size scales of the sem figures are not observed correctly. It is recommended to modify the figures so that they can be correctly observed.

Answer: The suggestion was taken into consideration and the text was modified accordingly.

Q4. To improve the identification of the results shown, it would be interesting to include in the titles of each of the sections whether we are working with the film or with the by-product. At the moment they are differentiated by numbering, but this could improve their identification.

Answer:  The suggestion was taken into consideration and the text was modified accordingly.

Q5. It is proposed that there is a difference between the different film thicknesses obtained as a function of particle size. But the reason for the change is not discussed.

Answer: The suggestion was taken into consideration and the text was modified accordingly.

Q6. In different sections it is proposed that there are different parameters that affect the mechanical properties, but they are not measured in this work. It would be interesting to incorporate this study as it can provide relevant information for its application.

Answer: Mechanical properties are of course very important, but not crucial at this stage, where only the ability to form films (per se) was tested, being the main focus of the article on the cellulose extraction methodologies. Further optimization of the films for the final application in heritage is going to be performed and published elsewhere, and mechanical properties are duly tested.

Reviewer 3 Report

Reviewer comments are attached. 

Reviewer comments are attached. 

Author Response

The authors would like to thank the reviewers for the suggestions made, we have carefully replied to all reviewer’s comments as seen below. All changes in the manuscript text have been highlighted in green.

  1. Please include the numerical values of results in the abstract to know the significance of the study performed.

Answer: The suggestion was taken into consideration and the text was modified to show the film’s thickness

  1. Introduction needs to be revised. The authors are requested to explain a comprehensive discussion about the current study. The present form of discussion is not satisfactory.

Answer: The suggestion was taken into consideration and the text was modified to better understand the current study.

  1. In Section 2.1, the major properties of the materials should be provided using a table.

 Answer: The authors were not able to find any published work with that information.

  1. The FT-IR spectra comparison between pure vine stalks, extracted cellulose, and the commercial controls is insightful. Please elaborate on how the differences in spectral peaks correspond to the specific chemical components in each sample?

Answer: The suggestion was taken into consideration and the text was modified accordingly.

  1. In your analysis of the extracted cellulose spectrum, you mentioned peaks at 889, 1024, 1200, 1306, and 3327 cm-1. Please discuss the significance of each of these peaks in the context of cellulose structure and functionality?

Answer: The suggestion was taken into consideration and the text was modified accordingly.

  1. The peaks at 889, 1024, 1200, 1306, and 3327 cm-1 suggest changes in the cellulose structure after treatment. Please provide insight into how these changes might affect the cellulose's behavior in different applications, such as its mechanical properties or interactions with other materials?

Answer: The suggestion was taken into consideration and the text was modified accordingly.

  1. Please provide insights into how the different thicknesses of the films (500 μm, 300 μm, 150 μm, and <150 μm) were chosen? Were there specific considerations or objectives that influenced these choices?

Answer: The suggestion was taken into consideration and an explanation was added to the introduction the explain the use of these sizes.

  1. The assertion that thickness control is critical for validating comparisons between films is intriguing. Please provide further insight into how variations in thickness could potentially impact the validity of mechanical property comparisons?

Answer: The suggestion was taken into consideration and the text was modified to show mechanical properties.

  1. The dominant endothermic peak observed between 100 °C and 146 °C is attributed to moisture evaporation. Please provide further insights into how the moisture content of cellulose films and granulates impacts their thermal behavior? How might this moisture content influence other properties of these materials?

Answer: The suggestion was taken into consideration and the text was modified to show the relevance of moisture content.

  1. The presence of a second endothermic peak attributed to cellulose decomposition is mentioned for some samples. Please discuss the factors that might contribute to this secondary peak and the implications of its presence in terms of the stability and thermal behavior of the cellulose films and granulates?

Answer: The suggestion was taken into consideration and the text was modified accordingly.

Round 2

Reviewer 2 Report

The article can be accepted in its present form, the authors have made the requested modifications.